# Fast tensor disentangling algorithm

**Kevin Slagle**[1,2★]

**1** Walter Burke Institute for Theoretical Physics,
California Institute of Technology, Pasadena, California 91125, USA
**2** Institute for Quantum Information and Matter,
California Institute of Technology, Pasadena, California 91125, USA

★ kslagle@caltech.edu

## Abstract

Many recent tensor network algorithms apply unitary operators to parts of a tensor network in order to reduce entanglement. However, many of the previously used iterative algorithms to minimize entanglement can be slow. We introduce an approximate, fast, and simple algorithm to optimize disentangling unitary tensors. Our algorithm is asymptotically faster than previous iterative algorithms and often results in a residual entanglement entropy that is within 10 to 40% of the minimum. For certain input tensors, our algorithm returns an optimal solution. When disentangling order-4 tensors with equal bond dimensions, our algorithm achieves an entanglement spectrum where nearly half of the singular values are zero. We further validate our algorithm by showing that it can efficiently disentangle random 1D states of qubits.



## Contents

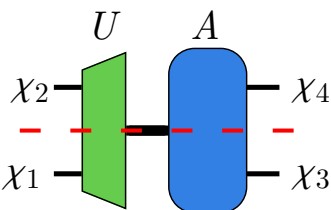

Figure 1: Given a tensor $A_{k,ab}$ (blue) with dimensions $(\chi_1\chi_2)\times\chi_3\times\chi_4$, where $\chi_1 \leq \chi_3$ and $\chi_2 \leq \chi_4$, our algorithm outputs a unitary tensor $U_{ij,k}$ (green) that roughly minimizes the entanglement across the red line.

# 1 Introduction

Many recent tensor network algorithms [1–7] rely on the application of unitary (or isometry) tensors in order too reduce the short-ranged entanglement and correlations within the tensor network. Examples of such algorithms include: MERA [8–10], Tensor Network Renormalization [11–13], Isometric Tensor Networks [14] and 2D DMRG-like canonical PEPS algorithms [15,16], purified mixed-state MPS [17], and unitary tensor networks [18–20]. Optimizing these unitary tensors is a difficult task, and many of the algorithms applied in the previously cited literature are CPU intensive, although there has been recent progress [20–22].

One popular approach is to optimize one tensor at a time while holding other tensors constant [10, 11, 15, 16, 20, 23, 24]. However, convergence can be slow, especially when a good initial guess for $U$ is not used.

Another approach (used in Refs. [14, 17]) is to iteratively minimize the entanglement entropy [defined later in Eqs. (10)] across part of the tensor network, as depicted in Fig. 1. However, these iterative methods are also very CPU intensive. An iterative first-order gradient descent algorithm can converge very slowly, especially when narrow valleys are present in the entanglement entropy cost function [1]. Convergence is even more challenging for Renyi entropies $S_\alpha$ with $\alpha \leq 1/2$ due to $|\lambda|^{2\alpha}$ singularities for small singular values $\lambda$, which can even prevent convergence to a local minima in the limit of infinitesimal step size for many algorithms. Applying a second-order Newton method can require less iterations. However, for large bond dimensions $\chi$ the CPU time and memory requirements grow rapidly as $O(\chi^{12})$ and $O(\chi^8)$, respectively, with e.g. $\chi = 16$ requiring roughly 40 CPU core hours and 34 GB of RAM just to diagonalize and store the Hessian for a single iteration.

In this work, we introduce a simple and asymptotically faster algorithm to calculate a reasonably good disentangling unitary tensor. That is, given a $(\chi_1\chi_2)\times\chi_3\times\chi_4$ tensor (blue in Fig. 1) with three indices where $\chi_1 \leq \chi_3$ and $\chi_2 \leq \chi_4$,[2] we provide an algorithm to efficiently calculate a $\chi_1 \times \chi_2 \times (\chi_1\chi_2)$ unitary[3] tensor (green) such that the entanglement is roughly minimized across the dotted red line.

The CPU time of our algorithm scales as $O(\chi_1^3\chi_3^2 + \chi_1^6)$ when $\chi_1 = \chi_2$ and $\chi_3 = \chi_4$.[4] This CPU complexity is as fast or faster (when $\chi_3 \gg \chi_1$) than the complexity $O(\chi_1^3\chi_3^3)$ for computing the singular values of $A$ across the dotted red line, which are needed to calculate the entanglement entropy across the dotted red line. This makes our algorithm asymptotically faster than just a single step of any iterative algorithm that attempts to minimize the entanglement entropy. For $\chi_1 = \chi_2 = \chi_3 = \chi_4 = 16$, our algorithm only requires only about 10ms of CPU time.

---

[1] Increasingly narrow valleys occur for the tensors in the later rows of Tab. 1

[2] In Appendix B, we generalize the algorithm to be applicable when $\chi_1 > \chi_3$ or $\chi_2 > \chi_4$.

[3] Here, unitary means that $\sum_k U_{ij,k}U^*_{i'j',k} = \delta_{ii'}\delta_{jj'}$ and $\sum_{ij} U_{ij,k}U^*_{ij,k'} = \delta_{kk'}$.

[4] See Appendix A for more complexity details.

We describe our algorithm in Sec. 2, and then benchmark it against iterative optimization of the entanglement entropy in Sec. 3. In Sec. 4, we show that our algorithm is capable of efficiently disentangling random initial states.

## 2 Algorithm and Intuition

---

**Algorithm 1:** Fast tensor disentangling algorithm [25]

**Input:** tensor $A_{k,ab}$ with dimensions $(\chi_1 \chi_2) \times \chi_3 \times \chi_4$, where $\chi_1 \leq \chi_3$ and $\chi_2 \leq \chi_4$
**Output:** unitary tensor $U_{ij,k}$ with dimensions $\chi_1 \times \chi_2 \times (\chi_1 \chi_2)$

1   $r_k \leftarrow$ random vector of length $\chi_1 \chi_2$
2   $\alpha_a^{(3)*}, \alpha_b^{(4)} \leftarrow$ dominant left and right singular vectors [5] of $(r \cdot A)_{ab}$
3   $V_{ai}^{(3)} \leftarrow$ from truncated SVD $\sum_b A_{k,ab} \alpha_b^{(4)} \approx (U^{(3)} \cdot \Lambda^{(3)} \cdot V^{(3)\dagger})_{ka}$,
    where $V^{(3)}$ is a $\chi_3 \times \chi_1$ semi-unitary
4   $V_{bj}^{(4)} \leftarrow$ from truncated SVD $\sum_a A_{k,ab} \alpha_a^{(3)} \approx (U^{(4)} \cdot \Lambda^{(4)} \cdot V^{(4)\dagger})_{kb}$,
    where $V^{(4)}$ is a $\chi_4 \times \chi_2$ semi-unitary
5   $B_{k,ij} \leftarrow \sum_{ab} A_{k,ab} V_{ai}^{(3)} V_{bj}^{(4)}$
6   $U_{ij,k} \leftarrow$ Gram-Schmidt orthonormalization of $(B^\dagger)_{ij,k}$,
        with $(i,j)$ grouped via the ordering described in main text

---

The algorithm is summarized in Algorithm 1. Below, we explain the algorithm in detail along with the underlying intuition.

To gain intuition, we will consider a simple example where the input tensor $A_{k,ab}$ is just a tensor product of three matrices:

$$A_{k,ab} \sim M_{k_1 a_1}^{(1)} M_{k_2 b_2}^{(2)} M_{a_2 b_1}^{(3)} \tag{1}$$

where we are grouping the indices $k = (k_1, k_2)$ and similar for $a$ and $b$. Then it is clear that an ideal unitary $U_{ij,k}$ should decompose $A_{k,ab}$ as follows:

$$(U \cdot A)_{ij,ab} \sim M_{ia_1}^{(1)} M_{jb_2}^{(2)} M_{a_2 b_1}^{(3)} \tag{2}$$

since this minimizes the entanglement across the cut shown in Fig. 1.

Note that $U_{ij,k}$ does not have any dependence on $M^{(1)}$, $M^{(2)}$, or $M^{(3)}$. Rather, $U_{ij,k}$ only needs to be a basis that matches the index $i$ with $M^{(1)}$ and $j$ with $M^{(2)}$. The indices $a$ and $b$ give us a handle on this basis since $M_{k_1 a_1}^{(1)}$ only depends on $a$ (and not $b$), and similar for $M_{k_2 b_2}^{(2)}$. However, the desired basis is obscured by $M_{a_2 b_1}^{(3)}$, which also depends on $a$ and $b$. Therefore, the intuition behind our algorithm will be to project out $M^{(3)}$ so that $U_{ij,k}$ can be computed.

Step **(1)** of the algorithm begins by choosing a random vector $r_k$ of length $\chi_1\chi_2$. **(2)** Then compute $\alpha_a^{(3)*}$ and $\alpha_b^{(4)}$: the dominant left and right singular vectors [5] of $(r \cdot A)_{ab}$.

For the simple example, $r \cdot A$ will be a tensor product of two matrices: $M^{(3)}$ and $L_{a_1b_2} = \sum_{k_1k_2} M_{k_1a_1}^{(1)} r_{k_1k_2} M_{k_2b_2}^{(2)}$. This implies that $\alpha_a^{(3)}$ and $\alpha_b^{(4)}$ will each be a tensor product of two vectors:

$$\alpha_{(a_1a_2)}^{(3)}, = \beta_{a_1}^{(3)}\gamma_{a_2}^{(3)}, \qquad\qquad \alpha_{(b_1b_2)}^{(4)}, = \gamma_{b_1}^{(4)}\beta_{b_2}^{(4)}, \tag{3}$$

where $\beta_{a_1}^{(3)*}$ and $\beta_{b_2}^{(4)}$ (and $\gamma_{a_2}^{(3)*}$ and $\gamma_{b_1}^{(4)}$) are the dominant left and right singular vectors of $L_{a_1b_2}$ (and $M_{a_2b_1}^{(3)}$), respectively. This allows us to isolate $M_{k_1a_1}^{(1)}$ by multiplying $A_{k,ab}$ by $\alpha_b^{(4)}$:

$$\sum_b A_{k,ab}\, \alpha_b^{(4)} \sim \sum_{b_1b_2} \left[ M_{k_1a_1}^{(1)} M_{k_2b_2}^{(2)} M_{a_2b_1}^{(3)} \right]\left[ \gamma_{b_1}^{(4)}\beta_{b_2}^{(4)} \right] \tag{4}$$

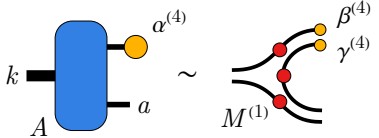

**(3-4)** Calculate the following truncated SVD [5]:

$$\sum_b A_{k,ab}\, \alpha_b^{(4)} \approx (U^{(3)} \cdot \Lambda^{(3)} \cdot V^{(3)\dagger})_{ka}, \tag{5}$$

$$\sum_a A_{k,ab}\, \alpha_a^{(3)} \approx (U^{(4)} \cdot \Lambda^{(4)} \cdot V^{(4)\dagger})_{kb}, \tag{6}$$

where only the largest $\chi_1$ and $\chi_2$ singular values are kept in the first and second lines, respectively. Thus, $V^{(3)}$ and $V^{(4)}$ are $\chi_3 \times \chi_1$ and $\chi_4 \times \chi_2$ semi-unitary matrices (i.e. $V^{(3)\dagger} \cdot V^{(3)} = \mathbb{1}$).

For the simple example, the matrices $V^{(3)}$ and $V^{(4)}$ only depend on the thin SVD of $M^{(1)} = U^{(1)} \cdot \Lambda^{(1)} \cdot V^{(1)\dagger}$ and $M^{(2)} = U^{(2)} \cdot \Lambda^{(2)} \cdot V^{(2)\dagger}$:

$$V_{(a_1a_2),i}^{(3)} = V_{a_1,i}^{(1)}\gamma_{a_2}^{(3)}, \qquad\qquad V_{(b_1b_2),j}^{(4)} = V_{b_2,i}^{(2)}\gamma_{b_1}^{(4)} \tag{7}$$

up to an unimportant tensor product with a vector $\gamma^{(3)}$ or $\gamma^{(4)}$. This allows us to project out $M^{(3)}$ in the following step, as seen in the bottom right of Eq. (8).

**(5)** Compute:

$$B_{k,ij} = \sum_{ab} A_{k,ab} V_{ai}^{(3)} V_{bj}^{(4)} \tag{8}$$

**(6)** Let $U_{ij,k}$ be the Gram-Schmidt orthonormalization of the rows of $(B^\dagger)_{ij,k}$.[6] If $\chi_1 \le \chi_2$, then the $(i,j)$ indices should be grouped via the ordering $(i,j) \to \chi_2 i + j$ so that $(U \cdot B)_{ij,i'j'} = 0$ if

---

[5]The singular value decompositions of a matrix $M$ takes the form $M = U\Lambda V^\dagger$, where $\Lambda$ is a diagonal matrix of singular values in decreasing order. The columns of $U$ and $V$ are the left and right singular vectors, respectively. The first column of $U$ and $V$ are the dominant left and right singular vectors. The truncated SVD results from keeping only the first $\chi$ columns of $U$ and $V$, and only the first $\chi$ rows and columns of $\Lambda$.

[6]If the rows (indexed by $i, j$) of $(B^\dagger)_{ij,k}$ are not linearly independent, then the remaining orthonormal vectors can be chosen randomly.

$\chi_2 i + j > \chi_2 i' + j'$, else the ordering $(i, j) \rightarrow \chi_1 j + i$ should be applied so that $(U \cdot B)_{ij,i'j'} = 0$ if $\chi_1 j + i > \chi_1 j' + i'$.[7]

For the simple example, due to the direct product structure of $V^{(3)}$ and $V^{(4)}$ shown in Eq. (7), $B_{k,ij}$ takes the form shown in the bottom right of Eq. (8). Importantly, $M^{(3)}$ only affects $B_{k,ij}$ by a multiplicative constant so that the indices $i$ and $j$ give us a good handle on how to split the index $k$. If $B_{k,ij}$ were unitary, which would be the case if $M^{(1)}$ and $M^{(2)}$ are unitary, then we could take $U = B^\dagger$ to minimize the entanglement across the cut as in Eq. (2). Since $B_{k,ij}$ is generally not unitary, we instead use a Gram-Schmidt orthonormalization of $(B^\dagger)_{ij,k}$. This produces the desired result [Eq. (2)] for the simple example (up to trivial multiplication of unitary matrices on $i$ and $j$ of $U_{ij,k}$).

Without loss of generality, let $\chi_1 \leq \chi_2$. Gram-Schmidt orthonormalization has the advantage that (as previously mentioned) $(U \cdot B)_{ij,i'j'} = 0$ if $\chi_2 i + j > \chi_2 i' + j'$, which results in at least $\frac{1}{2}\chi_1(\chi_1 - 1)$ zero singular values of $\widetilde{B}_{ii',jj'} = (U \cdot B)_{ij,i'j'}$ due to $\frac{1}{2}\chi_1(\chi_1 - 1)$ rows of zeroes in the matrix $\widetilde{B}$. When $\chi_1 = \chi_3$ and $\chi_2 = \chi_4$, $V^{(3)}$ and $V^{(4)}$ are unitary, and therefore $U \cdot A$ also has at least $\frac{1}{2}\chi_1(\chi_1 - 1)$ zero singular values (i.e. nearly half of the total $\chi_1^2$ singular values). More generally, $U \cdot A$ has at least

$$\frac{1}{2}\chi_1(\chi_1 - 1) - \max(\chi_1\chi_3, \chi_2\chi_4) + \chi_2^2, \qquad \text{for } \chi_1 \leq \chi_2 \qquad (9)$$

zero singular values [out of $\min(\chi_1\chi_3, \chi_2\chi_4)$] across the cut in Fig. 1. Note that Eq. (9) applies to general tensors $A_{k,ab}$, i.e. not just the particular form in Eq. (1).

Eq. (9) can be understood by defining $\widetilde{V}^{(3)}$ and $\widetilde{V}^{(4)}$ as any unitary matrices with $\widetilde{V}_{ai}^{(3)} = V_{ai}^{(3)}$ for $i \leq a$ and $\widetilde{V}_{bj}^{(4)} = V_{bj}^{(4)}$ for $j \leq b$. Note that $\widetilde{V}^{(3)}$ is a $\chi_3 \times \chi_3$ unitary while $V^{(3)}$ is a $\chi_3 \times \chi_1$ semi-unitary. Then $\widetilde{A}_{ii',jj'} = U_{ij,k} A_{k,ab} \widetilde{V}_{ai'}^{(3)} \widetilde{V}_{bj'}^{(4)}$ is a $\chi_1\chi_3 \times \chi_2\chi_4$ matrix with $\frac{1}{2}\chi_1(\chi_1 - 1)$ rows that each have $\chi_2^2$ out of $\chi_2\chi_4$ entries equal to zero. Since the rows are not entirely zero, there will be $\chi_2\chi_4 - \chi_2^2$ less zero singular values than than $\frac{1}{2}\chi_1(\chi_1 - 1)$. Furthermore, if $\chi_1\chi_3 > \chi_2\chi_4$, then $\widetilde{A}$ has more rows than columns, which will further decrease the number of zero singular values by $\chi_1\chi_3 - \chi_2\chi_4$, resulting in Eq. (9).

If $\chi_1 = \chi_3$ and $\chi_2 = \chi_4$, then $V^{(3)}$ and $V^{(4)}$ in Eq. (8) are just unitary matrices. Therefore $V^{(3)}$ and $V^{(4)}$ only change the basis of vectors that are Gram-Schmidt orthogonalized in step 6. One could then consider skipping steps 1-5 and instead input $B_{k,ij} = A_{k,ij}$ to step 6. The ansatz in Eq. (1) would still be optimally disentangled in this case. However, since the output of Gram-Schmidt depends on the initial basis, the resulting disentangling unitary will be different in general. Indeed, the resulting disentangling unitary will typically be significantly worse for general input tensors $A_{k,ij}$.[8]

The algorithm is not deterministic since $r_k$ is random, which helps guarantee the tensor product structure in Eq. (3) by splitting possibly degenerate singular values. Thus, it could be useful to run the algorithm multiple times and select the best result. Also note that the (statistical) result of the algorithm is not affected if $A$ is multiplied by a unitary matrix on any of its three indices. As such, it is not useful to rerun the algorithm on $U \cdot A$ (rather than just $A$) in an attempt to improve the result.

---

[7]In some cases, it could be useful to try both orderings and return the best resulting unitary.

[8]For the $\chi_1 = \chi_2 = \chi_3 = \chi_4 = 2$ tensors that we consider in Tab. 1, skipping steps 1-5 results in an entanglement $S^{\text{fast}}$ that is about 20 to 35% larger (on average).

# 3  Performance

Throughout this section, we assume $\chi_1 = \chi_2$ and $\chi_3 = \chi_4$. In Tab. 1, we show how well our algorithm minimizes the Von Neumann entanglement entropy:

$$S = -\sum_i p_i \log p_i, \qquad \text{where} \qquad p_i = \frac{\lambda_i^2}{\sum_j \lambda_j^2}, \qquad (10)$$

where $\lambda_i$ are the singular values of $U \cdot A$ across the red line in Fig. 1 [i.e. singular values of $(U \cdot A)_{ij,ab}$ when viewed as a $(\chi_1 \chi_3) \times (\chi_1 \chi_3)$ matrix with indices $(ia)$ and $(jb)$]. We also investigate the truncation error that results from only keeping the first $\chi$ singular values:

$$\epsilon_\chi = \sum_{i=\chi+1}^{\chi_1 \chi_3} p_i^2. \qquad (11)$$

In the first four rows, we investigate random $\chi_1^2 \times \chi_3 \times \chi_3$ tensors of complex Gaussian random numbers. We then consider random tensors with fixed singular values $\lambda_i = 1/i$ or $\lambda_i = 2^{-i}$, which are generated using

$$A_{(k_1 k_2),ab} = \sum_{i=1}^{\chi_1^2} \lambda_i W_{k_1 a,i} V_{k_2 b,i}, \qquad (12)$$

where $W$ and $V$ are random unitaries (e.g. $\sum_{k_1 a} W_{k_1 a,i} W_{k_1 a,j}^* = \delta_{ij}$). In the final three rows, we generate tensors using

$$A_{(k_1 k_2),ab} = \sum_{i=1}^{\chi_1^2} \mu_i v_{k_1}^{(1)} v_{k_2}^{(2)} v_a^{(3)} v_b^{(4)}, \qquad (13)$$

where $v^{(n)}$ are normalized random complex vectors. The later types of tensors have more structure and are (in a sense) less dense than the previous types.

We find that our fast algorithm performs best for more structured tensors (lower rows in the table) and exhibits the greatest speed advantages for larger $\chi$ and more structured tensors. The fast algorithm typically results in an entanglement $S^{\text{fast}}$ within 10 to 40% of the global minimum $S^{\text{min}}$ (which we approximate by running a gradient descent algorithm on several different initial unitaries for each input tensor). In the 5th column of Tab. 1, we show how much longer (on average) it takes the gradient descent algorithm to optimize down to the entanglement $S^{\text{fast}}$ reached by our fast algorithm; we find speedups ranging from 20 to 20,000 times as the bond dimension is increased from 2 to 16.

In the final two columns, we find that our fast algorithm achieves a truncation error to bond dimension $\chi_1$ that is within a factor of two of what is obtained by minimizing $S_1$. In Fig. 2, we study the truncation error in more detail. We find that if we truncate to a large enough bond dimension, our fast algorithm achieves a smaller truncation error than what is obtained by minimizing $S_1$. Both algorithms greatly reduce the truncation error from original random tensor $A$ (which we reinterpret as a tensor with four indices instead of three).

# 4  Wavefunction Disentangle

We further validate our algorithm by studying how well it can disentangle a random wavefunction of 10 qubits. [9] That is, starting from a wavefunction of $2^{10}$ complex Gaussian-distributed

---

[9]See Ref. [33] for an MPS approach to wavefunction disentangling.

Table 1: For various kinds of tensors $A$ [defined in Eqs. (12)-(13)], where $U \cdot A$ has dimensions $\chi_1 \times \chi_1 \times \chi_3 \times \chi_3$, we list: rough CPU time of our fast algorithm; roughly how much faster this is (on average) than a gradient descent algorithm of the entanglement entropy that halts at the entanglement $S^{\text{fast}}$ reached by our fast algorithm; how close the resulting entanglement entropy $S^{\text{fast}}$ of our fast algorithm is compared to the minimum $S^{\text{min}}$; the same but for a random unitary (for comparison); the truncation error $\epsilon^{\text{fast}}_{\chi_1}$ [Eq. (11)] to bond dimension $\chi_1$ for our fast algorithm; and the same for the unitary that minimizes the entanglement to $S^{\text{min}}$ (for comparison). The last four columns show means and the two-sided deviations to the 16th and 84th quantiles (e.g. a normal distribution would be shown as $\mu^{+\sigma}_{-\sigma}$.)

| tensor | $\chi_1$ | $\chi_3$ | time | speedup | $\frac{S^{\text{fast}}}{S^{\text{min}}}-1$ | $\frac{S^{\text{rand}}}{S^{\text{min}}}-1$ | $\epsilon^{\text{fast}}_{\chi_1}$ | $\epsilon^{\min S_1}_{\chi_1}$ |
|---|---|---|---|---|---|---|---|---|
| random | 2 | 2 | 0.5ms | 30x | $24^{+15}_{-10}\%$ | $130^{+80}_{-40}\%$ | $2.8^{+3}_{-2}\%$ | $1.0^{+1}_{-.5}\%$ |
| random | 2 | 4 | 0.5ms | 20x | $8^{+4}_{-3}\%$ | $17^{+5}_{-5}\%$ | $32^{+4}_{-4}\%$ | $27^{+3}_{-3}\%$ |
| random | 4 | 4 | 0.5ms | 100x | $34^{+5}_{-4}\%$ | $121^{+11}_{-11}\%$ | $13^{+2}_{-1}\%$ | $7.3^{+.7}_{-.7}\%$ |
| random | 16 | 16 | 10ms | 2000x | $50^{+.2}_{-.3}\%$ | $155^{+.8}_{-.5}\%$ | $36^{+.2}_{-.2}\%$ | $22^{+.1}_{-.04}\%$ |
| $\lambda_i = 1/i$ | 2 | 2 | 0.5ms | 35x | $22^{+15}_{-10}\%$ | $150^{+90}_{-40}\%$ | $2.4^{+2}_{-1}\%$ | $1.1^{+1}_{-.5}\%$ |
| $\lambda_i = 1/i$ | 4 | 4 | 0.5ms | 140x | $34^{+8}_{-7}\%$ | $203^{+20}_{-18}\%$ | $6.4^{+.9}_{-.8}\%$ | $4.1^{+.5}_{-.5}\%$ |
| $\lambda_i = 1/i$ | 16 | 16 | 10ms | 2000x | $57^{+2}_{-2}\%$ | $373^{+4}_{-5}\%$ | $11^{+.2}_{-.2}\%$ | $4.8^{+.05}_{-.05}\%$ |
| $\lambda_i = 2^{-i}$ | 2 | 2 | 0.5ms | 40x | $24^{+20}_{-10}\%$ | $260^{+90}_{-50}\%$ | $1.9^{+2}_{-1}\%$ | $0.9^{+1}_{-.4}\%$ |
| $\lambda_i = 2^{-i}$ | 4 | 4 | 0.5ms | 160x | $43^{+14}_{-10}\%$ | $320^{+50}_{-40}\%$ | $3.0^{+.7}_{-.6}\%$ | $1.4^{+.2}_{-.2}\%$ |
| $\lambda_i = 2^{-i}$ | 16 | 16 | 10ms | 3000x | $77^{+3}_{-4}\%$ | $735^{+23}_{-18}\%$ | $2.7^{+.1}_{-.1}\%$ | $0.5^{+.01}_{-.01}\%$ |
| $\mu_i = 1/i$ | 2 | 2 | 0.5ms | 120x | $30^{+30}_{-8}\%$ | $200^{+150}_{-70}\%$ | $1.0^{+1}_{-.5}\%$ | $0.3^{+.4}_{-.1}\%$ |
| $\mu_i = 1/i$ | 4 | 4 | 0.5ms | 300x | $19^{+8}_{-5}\%$ | $139^{+26}_{-17}\%$ | $3.0^{+.8}_{-.6}\%$ | $1.5^{+.4}_{-.3}\%$ |
| $\mu_i = 1/i$ | 16 | 16 | 10ms | 20,000x | $8^{+3}_{-1}\%$ | $161^{+5}_{-2}\%$ | $3.0^{+.1}_{-.1}\%$ | $1.7^{+.1}_{-.1}\%$ |

random numbers, we repeated apply our algorithm to different parts of the wavefunction, see inset of Fig. 3, to reduce the amount of entanglement across any cut of the wavefunction. Thus, we take $A_{(k_i,k_{i+1}),(a_1 \cdots a_{i-1})(b_{i+2} \cdots b_n)} = \psi_{a_1 \cdots a_{i-1} k_i k_{i+1} b_{i+2} \cdots b_n}$ in Fig. 1 for $i = 1, 3, \ldots, n-1$ and then $i = 2, 4, \ldots, n-2$ to calculate the two layers of unitaries shown in the inset of Fig. 3, for which $n = 10$.[10] We show how much entanglement is left after a given number of layers of unitaries. We compare data from our fast disentangling algorithm to gradient descent of the entanglement entropy $S$ [Eq. (10)].

When the circuit depth is small, our fast algorithm disentangles at a slightly slower rate per circuit layer, but much faster per CPU time. When the circuit depth is larger and the wavefunction has little entanglement left, our algorithm performs better than minimizing the entanglement entropy. Gradient descent of $S$ gets stuck at larger depth due to narrow valleys in the cost function $S$, which result in very small ($< 10^{-8}$) step sizes causing our gradient descent algorithm to halt.

We also compare against initializing the gradient descent of $S$ algorithm with the result of our fast disentangling algorithm. This is shown in blue in Fig. 3, and achieves the best disentangling rate in both limits, while also speeding up the gradient descent algorithm by a factor of two.

After 500 layers consisting of 2250 2-qubit gates, our fast algorithm removed almost all of the entanglement. An arbitrary 2-qubit gate can be implemented using three CNOT gates

---

[10]At the edges where $i = 1$ or $i = n-1$, we will not have $\chi_1 \leq \chi_3$ and $\chi_2 \leq \chi_4$. Therefore we use the extension in Appendix B for these two cases.

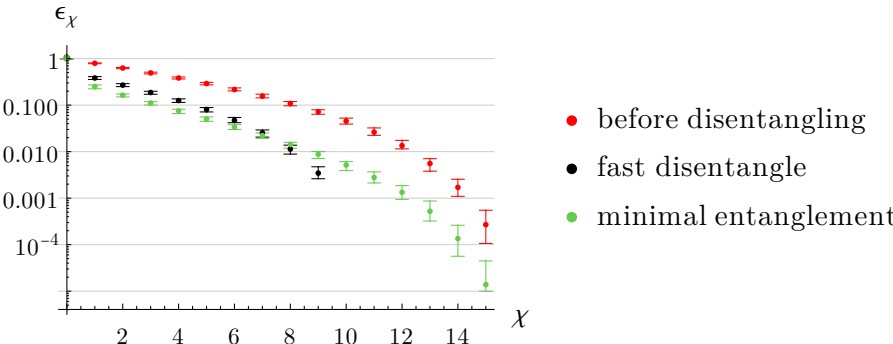

Figure 2: The truncation error [Eq. (11)] to bond dimension $\chi$ where $U \cdot A$ has dimensions $4 \times 4 \times 4 \times 4$ and the tensor $A$ is Gaussian random. Similar to Tab. 1, we also show the two-sided deviations to the 16$^{\text{th}}$ and 84$^{\text{th}}$ quantiles. We see that our fast algorithm outperforms the minimal entanglement disentangler for $\chi \geq 8$ and has zero truncation error for $\chi \geq 10$, which is consistent with the 6 zero singular values predicted by Eq. (9).

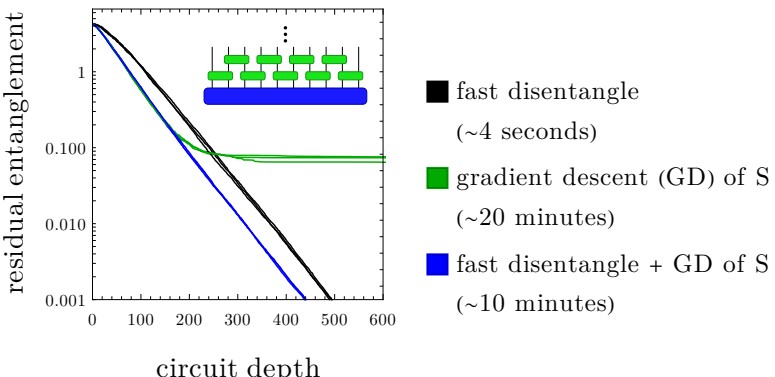

Figure 3: The residual entanglement after applying layers of unitary operators to a random 10-qubit wavefunction for various algorithms. The residual entanglement is the maximum entanglement across any left/right cut of the wavefuntion. We apply each method to the same three random wavefunctions, resulting in three nearly overlapping lines for each method. We also show the amount of CPU time used for each method to obtain the data shown. (inset) Two layers (i.e. depth=2) of unitaries acting on the wavefunction.

along with 1-qubit gates [26–28]. Therefore, the fast algorithm's circuit of 2250 2-qubit gates can be implemented using only 6750 CNOT gates. For comparison, it is possible to exactly disentangle an $n = 10$ qubit state using a circuit of $9 \times 2^{n+1} \approx 18,000$ nearest-neighbor 2-qubit CNOT gates along with many 1-qubit gates [29,30].

## 5 Conclusion

We have introduced, provided intuition for, and benchmarked a fast algorithm to approximately optimize disentangling unitary tensors. Example Python, Julia, and Mathematica code can be found at Ref. [25].

We expect our algorithm to be useful for tensor network methods that require disentangling unitary tensors. Due to its speed, our fast method can allow for simulating significantly larger bond dimensions than previously possible. The advantages of larger bond dimensions could outweigh the disadvantage of the non-optimal disentangling unitaries that our algorithm returns. Nevertheless, if more optimal unitaries are required, our fast algorithm can still be useful as a way to initialize an iterative algorithm.

For future work, it would be useful to consider an ansatz of tensors that are a tensor product of our ansatz Eq. (1) with a GHZ state (i.e. $A_{k,ab} = 1$ if $k = a = b$ else 0). Such tensors are the generic form of stabilizer states with three indices (up to unitary transformations on the three indices) [31,32]. Although these tensors can be optimally (and relatively easily) disentangled using a Clifford group unitary, our fast algorithm performs very poorly on these tensors.

## Acknowledgements

We thank Miles Stoudenmire and Michael Lindsey for helpful discussions and suggestions.

**Funding information** K.S. is supported by the Walter Burke Institute for Theoretical Physics at Caltech; and the U.S. Department of Energy, Office of Science, National Quantum Information Science Research Centers, Quantum Science Center.

## A CPU Complexity

The CPU complexity of our algorithm is

$$O\left[(\chi_1\chi_2)^2(\chi_3 + \chi_4) + \chi_1\chi_2\chi_3\chi_4 \min(\chi_1, \chi_2) + (\chi_1\chi_2)^3\right], \tag{14}$$

where we continue to assume $\chi_1 \leq \chi_3$ and $\chi_2 \leq \chi_4$. The first term results from steps 3, 4, and 5 in our Algorithm 1; the second term comes from step 5; and the final term results from step 6.[11] When $\chi_1 = \chi_2$ and $\chi_3 = \chi_4$, this reduces to $O(\chi_1^3\chi_3^2 + \chi_1^6)$.

---

[11]We assume that the dominant singular vectors of an $m \times n$ matrix $M$ can be calculated in time $O(mn)$ for step 2. This can be done for an $m \times n$ matrix (where $m \geq n$) with SVD $M_{m\times n} = U_{m\times n}\Lambda_{n\times n}V_{n\times n}^\dagger$ by e.g. applying the Lanczos algorithm to obtain the first singular vectors of $M^\dagger M = V\Lambda^2 V^\dagger$ and $MM^\dagger = U\Lambda^2 U^\dagger$, or $\begin{pmatrix} 0 & M \\ M^\dagger & 0 \end{pmatrix} = Y \begin{pmatrix} \Lambda & 0 \\ 0 & -\Lambda \end{pmatrix} Y^\dagger$ where $Y = \frac{1}{\sqrt{2}} \begin{pmatrix} U & U \\ V & -V \end{pmatrix}_{(m+n)\times 2n}$, for which the eigendecompositions reveal the SVD decomposition. For steps 3 and 4, we assume that the truncated SVD of an $m \times n$ with $m \leq n$ matrix that returns only the first $k \leq m$ singular vectors can be calculated in time $O(m^2 n)$. The final complexity in Eq. (14) would not change if the truncated SVD only required $O(mkn)$ time. The precision of these SVD steps is not critically important, and a fast $O(mkn)$ SVD method [34] can be safely applied if desired.

Remarkably, this is as fast or faster than computing a single SVD of $A$ [viewed as a $(\chi_1\chi_2) \times (\chi_3\chi_4)$ matrix] or even just computing $U \cdot A$, which both scale as $O[(\chi_1\chi_2)^2(\chi_3\chi_4)]$.

## B $\quad \chi_1 > \chi_3$

The algorithm can be extended to handle $\chi_1 > \chi_3$ as long as $\chi_2 \leq \chi_4'$, where we define

$$\chi_{4\to3} = \lceil \chi_1/\chi_3 \rceil, \qquad\qquad \chi_4' = \lceil \chi_4/\chi_{4\to3} \rceil. \tag{15}$$

$\lceil \chi_1/\chi_3 \rceil$ denotes the ceiling of $\chi_1/\chi_3$. If both fractions are integers, then $\chi_2 \leq \chi_4'$ is equivalent to $\chi_1\chi_2 \leq \chi_3\chi_4$. If instead $\chi_2 > \chi_4'$, then this appendix can be applied after swapping $\chi_1 \leftrightarrow \chi_2$ and transposing the last two indices of $A$. Therefore, this appendix extends our algorithm so that it can be applied as long as either $\chi_2 \leq \lceil \frac{\chi_4}{\lceil \chi_1/\chi_3 \rceil} \rceil$ or $\chi_1 \leq \lceil \frac{\chi_3}{\lceil \chi_2/\chi_4 \rceil} \rceil$ (although $\chi_1\chi_2 \leq \chi_3\chi_4$ is often sufficient[12]). This $\chi_1 > \chi_3$ case algorithm appears to result in an optimal disentangling unitary for the ansatz in Eq. (1) if one of the dimensions of $M^{(3)}$ is 1.

Suppose $\chi_1 > \chi_3$ and $\chi_2 \leq \chi_4'$. The algorithm proceeds as follows:

**(1)** Calculate the SVD:

$$A_{k,ab} = \sum_i U^{(1)}_{ka,i} \lambda^{(1)}_i V^{(1)*}_{bi}. \tag{16}$$

**(2)** Split the index $i$ of $V^{(1)}$ into two indices: $i \to (a', b')$ where $i = (a'-1)\chi_4' + b'$ and $1 \leq a' \leq \chi_{4\to3}$ and $1 \leq b' \leq \chi_4'$. If $\chi_4 < \chi_{4\to3}\chi_4'$, then append columns of zero vectors to $V^{(1)}$ before splitting the index. This results in a new $\chi_4 \times \chi_{4\to3} \times \chi_4'$ tensor $\widetilde{V}_{b,a'b'}$.

**(3)** Perform a (slightly modified) fast disentangling Algorithm 1 on

$$A'_{k,(aa')b'} = \sum_b A_{k,ab} \widetilde{V}_{b,a'b'}, \tag{17}$$

where the second index of $A'_{k,(aa')b'}$ is the grouped index $(a, a')$. The fast disentangling algorithm is modified in step 6: the $(i, j)$ indices should always be grouped using the ordering that would be chosen if $\chi_1 \leq \chi_2$.

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
