# Peer review of "Fast Tensor Disentangling Algorithm"

_SciPost Physics, doi:SciPost Phys. 11, 056 (2021)_

## Round 1 · Referee Report · Anonymous (Referee 1) · 2021-6-1

Report

The paper introduces and numerically demonstrates an algorithm that quickly generates disentangling unitaries for use in tensor network algorithms. The author also provides readable example Python code through GitHub. I recommend the work for publication in SciPost Physics: the work introduces a relevant new tool for operator disentangling, proves better asymptotic scaling than iterative algorithms in some regimes, and numerically demonstrates a significant speedup over a gradient descent algorithm.

In its base form, the "fast-disentangling algorithm" provides a single unitary $U$ that reduces the entanglement across a specified cut of an input four-index tensor $A$ by an a priori unknown amount. On the analytical side, equation 9 gives rigorous lower bounds for the number of Schmidt coefficients of $UA$ that are identically zero. The author further motivates the algorithm by showing it maximally reduces the entanglement for a restricted class of tensors given in equation 1.

The author demonstrates numerically that the algorithm is considerably faster than iterative methods by timing how long an iterative method (gradient descent) takes to reach the same reduction in entanglement entropy as the fast-disentangling algorithm. (The iterative algorithm may ultimately reduce the entanglement entropy more than the fast-disentangling algorithm, which, when straightforwardly applied, does not give improvements upon iteration.) The author also proves that the asymptotic scaling of the fast-disentangling algorithm is better than that of single steps of iterative algorithms. The asymptotic difference is largest for tensors $A$ with different dimension indices where the disentangling unitary is contracted with the two smallest dimensional indices of $A$.

Finally, the author demonstrates how to disentangle a wavefunction using a brickwork circuit generated by the fast-disentangling algorithm iterated over location of the entanglement cut. This example also shows that the fast-disentangling algorithm can be paired well with iterative ones for further reduction of entanglement entropy.

Overall, the algorithm is simple but powerful, the asymptotics follow straightforwardly from the algorithm, and the numerical demonstration is convincing. I recommend publication in SciPost Physics. There are a couple places where the clarity can be improved, and I request a couple changes and suggest several more. Please see the requested changes for more details.

Requested changes

Warnings issued while processing user-supplied markup:

  • Inconsistency: plain/Markdown and reStructuredText syntaxes are mixed. Markdown will be used.
    Add "#coerce:reST" or "#coerce:plain" as the first line of your text to force reStructuredText or no markup.
    You may also contact the helpdesk if the formatting is incorrect and you are unable to edit your text.
  1. What is the definition of $S^{min}$ in table 1? How does one compute it and the tensor corresponding to it? Please specify this in a few sentences in section 2.

2. Section 1 shows that the algorithm gives optimal results for entanglement minimization for the restricted form of the tensor $A$ given in equation 1. Equation 9, detailing the minimum number of singular values that are set identically zero, applies to more general tensors $A$ than just the restricted form given in equation 1. This difference should be explicitly reiterated/emphasized in the final paragraph of section 1 for clarity’s sake.

The following suggestions are optional.

  1. On the right hand side of equation 1, consider including a three-indexed unitary block (with shape [$\chi_1\chi_2$, $\chi_1$, $\chi_2$]) contracted with the left hand side of $M^{(1)}$ and $M^{(2)}$. That will make the difference between the right hand sides of equation 1 and equation 2 clearer. I think it will also help introduce the idea of "projecting out $M^{(3)}$" in order to isolate the piece of the tensor $A$ that is easily and optimally modified by left multiplication by a unitary.

  2. Assuming for ease that all the legs of $A$ have the same dimension, $\chi_1 = \chi_2=\chi_3=\chi_4$, are there drawbacks to skipping immediately to the final step, 6, of the algorithm, using $A$ in the place of $B$? Running that step alone will ensure the same minimum number of identically vanishing singular values. The drawback would be that the resulting algorithm would likely no longer be an optimal algorithm for tensors $A$ with forms similar to that given in equation 1. Nevertheless, might it give comparable results for the unstructured, random matrices? Including a declarative statement on this, or an additional column or two in table 1, will help further emphasize that steps 1-5 are important for fast disentangling of structured tensors and less important for unstructured ones.

  3. Section 3 discusses disentangling a wavefunction with the fast-disentangling algorithm. The procedure quickly, non-iteratively reduces the entanglement entropy across a given cut using the fast-disentangling algorithm, and then iterates over the location of the cut. Additional iteration is then possible by applying gradient descent algorithms to the resulting brickwork circuit, or loading the resulting state as the initial state of another iterative algorithm. Even without those additional iterative procedures, this application of the fast-disentangling algorithm is iterative given the iteration over the location of the cut and the generation of many layers of a brickwork circuit. Please comment in a sentence in section 3 how this is consistent with the following sentence from section 1: “The (statistical) result of the algorithm is not affected if $A$ is multiplied by a unitary matrix on any of its three indices. As such, [it] is not useful to iterate this algorithm.” From a little thought on the reader’s part, the consistency stems from the resulting brickwork layers multiplying more than a single index at once, but this should be explicitly spelled out.

  4. Does the iterative approach (generating a brickwork circuit by iterating over entanglement cuts) referenced in point 4 generalize? For example, for an order-4 tensor with $\chi_1 = \chi_2 = \chi_3 = \chi_4 = 16$, what of treating the $k$ index with dimension $\chi_1^2$ as a set of eight 2-dimensional indices and iterating over “local” pairs of them similarly to the wavefunction disentangler mentioned above? The new $\chi_1’$ would be $2$, and $\chi_3’=128$. The resulting scaling, $O(\chi_1^4 \chi_3 + \chi_1^6)$, would be in principle reduced by a factor of 10000 per unitary relative to one-off use of the fast-disentangling algorithm. The drawback might be that the small size of the disentangling unitaries and the fact that many do not directly reduce the entanglement across the original cut of interest would reduce the efficacy. Nevertheless, could the iterated fast-disentangling algorithm’s brickwork be “better” (in e.g. final entanglement entropy after some large number of iterations or time to reach a low entanglement entropy) than a one-off use of the fast-disentangling algorithm? Please briefly comment in a couple sentences on the applicability/future potential of the iterative approach beyond wavefunctions.

  • validity: top
  • significance: good
  • originality: high
  • clarity: good
  • formatting: good
  • grammar: excellent

Author:  Kevin Slagle  on 2021-06-25  [id 1523]

(in reply to Report 1 on 2021-06-01)
Category:
answer to question

Thank you for thoroughly reviewing our work and for the helpful suggestions.

  1. S^min is the smallest possible entanglement that can be achieved by a disentangling unitary. We now explain that the approximate S^min by running a gradient descent algorithm on several different initial unitaries for each input tensor. The 6th column of Table 1 is then the average (over different random tensors 'A') of S^fast/S^min - 1.

  2. Thank you for the suggestion. We emphasize now this below Eq. (9).

  3. That is a very interesting point! If $\chi_1 = \chi_3$ and $\chi_2 = \chi_4$, then $V^{(3)}$ and $V^{(4)}$ in Eq (8) are unitary matrices. Thus, $V^{(3)}$ and $V^{(4)}$ can only change the basis of vectors that are Gram–Schmidt orthogonalized. The ansatz in Eq. (1) will still be optimally disentangled. However, since the output of Gram–Schmidt depends on the initial basis, the resulting disentangling unitary will be different in general. We find that for the tensors considered in Table 1, skipping steps 1-5 will typically result in significantly worse disentangling unitaries, and about 20 to 35% more entanglement for the different kinds of $\chi_1 = \chi_2 = \chi_3 = \chi_4 = 2$ tensors in Table 1. We added this discussion near the end of Section 1.

  4. That is a good point. We reworded the last sentence of Section 1 in order to avoid this confusion.

  5. That is an interesting idea. However, our disentangling algorithm (and gradient descent) performs significantly worse when trying to disentangle an operator (rather than a wavefunction) using a brick circuit. For example, it should always be possible to disentangle a unitary operator by simply acting with small unitaries on one side of the operator. Unfortunately however, even for unitary operators, generating a brick circuit by locally minimizing entanglement (as we did in Section 3), does not disentangle the unitary to an identity operator (when using our fast algorithm or a gradient descent algorithm for each step). (Refs [33-34] discuss a different approach for generating a brick circuit.) Therefore, we do not expect splitting the $k$ index as described and iterating across it would perform well. However, we have not checked.

Anonymous on 2021-07-08  [id 1556]

(in reply to Kevin Slagle on 2021-06-25 [id 1523])

It is possible that the gradient descent algorithm's $S^\text{min}$ does not always reach the global minimum. However, we did make an effort to check that if that does occur, it does not occur frequently enough to significantly affect the numbers in Table 1.

The fact that the fast disentangling algorithm performs significantly worse on stabilizer states is not due to an inaccurate $S^\text{min}$. Stabilizer states are not handled well because the ansatz Eq 1 is (in a sense) maximally violated by stabilizer states (that include a GHZ state component). In contrast, the $\mu_i = 1/i$ tensors in Table 1 are much closer to the anstaz, which is why the algorithm performs better for those tensors.

Author:  Kevin Slagle  on 2021-06-30  [id 1535]

(in reply to Kevin Slagle on 2021-06-25 [id 1523])
Category:
correction

It seems SciPost automatically renumbered my replies. The points labeled 3-5 above should actually be labelled 4-6. We appreciate the optional suggestion 3, but chose not to implement it.

Anonymous on 2021-06-29  [id 1534]

(in reply to Kevin Slagle on 2021-06-25 [id 1523])
Category:
question

Is there any chance that the gradient descent algorithm's $S^{min}$ is not giving a good proxy for the true $S^{min}$? For example, you noted on the other review that the fast disentangling algorithm performs significantly worse on stabilizer states; could it merely be that the proxy $S^{min}$ for stabilizer states is more accurate (smaller)?

---

## Round 1 · Referee Report · Anonymous (Referee 2) · 2021-6-14

Report

In his manuscript, the author proposes a novel numerical algorithm, aimed at finding a unitary transformation that minimizes the entanglement across some particular set of legs of a multi-index tensor. Such a procedure is an important part of certain tensor network based algorithms, such as the multi-scale entanglement renormalization ansatz (MERA) and the related tensor network renormalization (TNR) procedure. It also has significance on its own: in the context of quantum computation, one is often interested in the minimal number of unitary gates needed to prepare a particular quantum state, known as the state's circuit complexity.

The paper clearly explains the proposed algorithm and provides evidence that it is significantly faster than previously existing methods, especially for large tensors, while it is almost as effective in minimizing the entanglement. While the benchmarking results presented are somewhat limited (i.e., they focus mostly on toy examples of various randomly chosen tensors, rather than using the method in the context of a particular algorithm such as TNR), based on them, it is reasonable to assume that the new 'fast disentangling' method will be a useful addition into the numerical arsenal of tensor network algorithms. For this reason, I think it is well suited for publication in SciPost.

One shortcoming of the paper is that, while there is some intuition provided for the algorithm, it is somewhat unclear what are the necessary conditions for it to work well. Are there examples of states where it fares much worse than previous disentangling methods? In the application presented in the paper, the algorithm is used to disentangle a completely random states, which is a 'worst case' scenario. One might wonder if the method still continues to outperform existing ones for states that are known to be easy to disentangle in principle. I think it would improve the manuscript if the author could comment on this question.

As a technical comment: when testing the algorithm, I have noticed a small issue. In the Schmidt orthonormalization step, when rows of the matrix are not exactly linearly dependent, only within machine precision, the algorithm fails to detect this and sometimes returns results that are not unitary. As far as I can see, this can be easily remedied by replacing the condition 'norm == 0' with some small cutoff instead.

Requested changes

I have found a small number of typos in the manuscript. 1) 'tensor project' (presumably should be 'tensor product') 2) In the line following footnote 3: chi1 -> chi_1

  • validity: -
  • significance: -
  • originality: -
  • clarity: -
  • formatting: -
  • grammar: -

Author:  Kevin Slagle  on 2021-06-25  [id 1522]

(in reply to Report 2 on 2021-06-14)
Category:
answer to question

Thank you for reviewing our work and especially for testing our algorithm.

Indeed, it is still unclear what the necessary conditions are for "good performance", as it is also unclear how to define "good performance". However, there is an important example where the fast algorithm performs significantly worse for certain easy to disentangle states: the stabilizer states. We now explain this more thoroughly in the last paragraph of the conclusion.

Thank you for the technical comment! We have recently significantly improved the robustness of the Python and Julia implementations in the GitHub such that this problem is avoided.

Thank you for pointing out those typos!

---

## Round 3 · Author Response

We thank the referees for the very helpful reviews of our work, which has led to useful improvements in our draft.

---

## Round 3 · List of Changes

A detailed diff showing changes can be found here:
https://drive.google.com/file/d/1qF-I1ceZBlpYFFnZdiYg1GB8X7mOkDwg/view?usp=sharing

In addition to the changes suggested by the referees, we improved the extended algorithm in Appendix B.

---

## Editorial Decision

published